# Use of a new micropattern tape method to detect chirality shifts in differentiating C2C12 cells

**Qingkai Weng** [1], **Takashi Osaka** [2], **Hiroaki Onoe** [3], **Koki Yoshida** [4], **Yukiko Kuroda** [1], **Koichi Matsuo** [1], **Katsuhiro Kawaai** [1]*

1 Laboratory of Cell and Tissue Biology, Keio University School of Medicine, Shinjuku-ku, Tokyo, Japan, 2 Tokyo Ohka Kogyo Co., Ltd., Kawasaki, Kanagawa, Japan, 3 Department of Mechanical Engineering, Faculty of Science and Technology, Keio University, Yokohama, Kanagawa, Japan, 4 Mechanical Engineering Program, College of Engineering, Shibaura Institute of Technology, Toyosu, Koto-ku, Tokyo, Japan

* k-kawaai@keio.jp

## Abstract

Chirality is an intrinsic property of cells manifested as left-right (LR) asymmetry in terms of cellular morphology and organization, which influences cell behavior, migration, and tissue development. Traditional *in vitro* methods used to study cell chirality often require complex fabrication methods, limiting their accessibility and reproducibility. Here, we present a novel micropattern tape method that facilitates fabrication of high-quality rectangular micropatterns useful for efficient, high-throughput analysis of cell chirality. Using this method, we characterized chirality of C2C12 myoblasts and MC3T3-E1 osteoblasts, which respectively exhibit clockwise (CW) and counterclockwise (CCW) chirality relative to the long axis of the rectangle. We used the method to analyze how cellular differentiation impacts chirality and observed striking reversal of C2C12 cell chirality upon bone morphogenic protein-2 (BMP2)-induced osteoblastic differentiation. These results demonstrate that our micropattern tape method can effectively detect dynamic change of cell chirality during differentiation.

## Introduction

Chirality is fundamentally defined as the inability of an object to be superimposed on its mirror image following spatial rotation and translation. Cell chirality is a ubiquitous phenomenon observed in both unicellular and multicellular organisms and manifested as intrinsic left-right (LR) asymmetry in cell morphology and intercellular organization [1,2]. Various approaches have been developed to detect and characterize cell chirality *in vitro*, including analysis of multicellular alignment [3], organelle positioning [4], actin filament self-organization [5], and focal adhesion localization using micropatterning [6]. Cell chirality has profound implications for cell behavior and migration, and for breaking LR asymmetry [7–10]. Cell chirality also functions to regulate

**Data availability statement:** All relevant data are within the paper and its Supporting Information files.

**Funding:** This research was supported by the JSPS KAKENHI grant number 21H05789, 21H03060 and 23K21467, and by the Keio Global Science Campus. The funders had no role in study design, data collection and analysis, decision to publish, or preparation of the manuscript.

**Competing interests:** No authors have competing interests. This does not alter our adherence to PLOS ONE policies on sharing data and materials.

intercellular junctions and endothelial permeability [11], and changes in chirality can serve as a potential indicator of cytotoxicity [12].

In this study, we developed a new micropattern tape that enables easier and more stable fabrication of high-quality micropatterns to allow analysis of cell chirality through efficient detection of dominant orientations. We validated this method by analyzing chirality of C2C12 myoblasts and MC3T3-E1 osteoblasts, two cell types of opposite chirality [13]. Treatment with bone morphogenetic protein-2 (BMP2) induces osteoblastic differentiation of C2C12 myoblasts [14]. These reports prompted us to compare the chirality of C2C12 cells before and after BMP2 treatment with that of MC3T3-E1 cells thereby revealing the possible effect of induced cell differentiation on cell chirality.

## Methods

### Cell culture

Mouse preosteoblast MC3T3-E1 cells were purchased from ATCC (Subclone 4, CRL-2593) and maintained in ascorbic acid-free αMEM (A1049001, Thermo Fisher) supplemented with 10% fetal bovine serum (FBS) and a 1% penicillin-streptomycin mixture (25253–84, Nacalai Tesque). Mouse myoblast C2C12 cells were purchased from ATCC (CRL-1772) and maintained in DMEM (08458–45, Nacalai Tesque) supplemented with 10% fetal bovine serum and 1% penicillin-streptomycin mixture. Every 3–4 days, the culture medium was changed, or cells were passaged. All cells were maintained in a humidified incubator at 37°C and 5% $CO_2$.

### Micropattern fabrication

Micropattern tape (custom-designed IHC-Fencing Seal, CS CRIE) was applied to an uncoated film-bottom dish (81151, ibidi), and surface hydrophilicity was induced by plasma activation treatment at soft mode for 90 seconds with a plasma ion bombarder (PIB-10, VACUUM DEVICE). Afterwards, the tape-film surface was washed with 95% ethanol to fill vacant areas and then washed 3 times with reverse osmosis (RO) water. After a wash with phosphate buffered saline (PBS), the surface (4.1 cm$^2$) was filled with 300 μL of 40 μg/mL fibronectin (FC010, Sigma-Aldrich) or 13 μg/mL vitronectin solution (V0132, Sigma-Aldrich), incubated 1 hour at room temperature (RT), washed 3 times with PBS and dried by aspiration. Micropattern tape was then removed, and the surface was treated 10 minutes with 0.5% Pluronic F-127 solution (59000, Biotium, Inc.) at RT to prevent cell adhesion to uncoated areas. After removing the Pluronic F-127 solution and washing 3 times with RO water, the surface was exposed to ultraviolet (UV) light for sterilization before use.

### Cell seeding on the micropattern

Seeding density of 5x10$^4$ cells/2 mL per dish was optimized using MC3T3-E1 cells, as described in Appendix S1. After trypsinization, cells were counted with a Cell Counter (Model R1, Olympus) and then seeded onto micropattern dishes. To guarantee even distribution of cells across micropatterns, dishes were gently moved back

and forth and side to side, approximately five times every 10 minutes, for about half an hour until cells adhered. This step prevented cells from concentrating in the center of the dish, which can result in uneven cell adhesion.

## Osteoblastic differentiation and evaluation of alkaline phosphatase activity

To assess osteoblastic differentiation, cells were seeded at a density of $2 \times 10^4$ cells/cm$^2$ on plastic dish (353230, Corning) coated with fibronectin (50 µg/mL, FC010, Merck Millipore), vitronectin (2.6 µg/mL, V0132, Sigma-Aldrich), or type I atero-collagen (500 µg/mL, IPC-30, KOKEN) by standard coating method (4˚C overnight and PBS wash). Non coating plastic dish was used for experiments for evaluating cell culture medium and control experiment to evaluate coating proteins. C2C12 cells were cultured with growth medium containing 100 or 300 ng/mL bone morphogenetic protein-2 (BMP2) (355-BM-050, R&D SYSTEMS) for osteoblastic differentiation. On day 4 the medium was refreshed, and after 6 days of culture, cells were fixed 10 minutes in 4% paraformaldehyde (PFA)/PBS, permeabilized for 5 minutes in 0.5% Triton-X100, and evaluated for alkaline phosphatase (ALP) activity with an ALP staining kit (85L1-1KT, Sigma-Aldrich). To evaluate myosin heavy chain (MHC) expression and ALP activity simultaneously, the cells were first fixed with 4% PFA/PBS and then permeabilized with 0.2% Triton X-100. Next, the cells were blocked with a blocking buffer containing 5% normal donkey serum (D9663, Sigma-Aldrich), 10 µg/mL donkey IgG (017-000-003, Jackson ImmunoResearch Laboratories), and 1% bovine serum albumin (BAC62, Equitech-Bio). The cells were stained with anti-MHC antibody (MF20, DSHB) for two hours, washed with PBS, and stained with anti-mouse-HRP antibody (715-035-151, Jackson ImmunoResearch Laboratories) for one hour. After washing with PBS, ALP staining was performed (85L1-1KT, 37˚C, 30 min). The cells were washed with water and stained with a DAB staining kit (Immpact DAB kit, SK-4105, Vector Laboratories) to detect the HRP antibody. Photographs of stained cells were obtained using Thunder Imaging Systems (Leica), and ALP-positive area and HRP-positive area were quantified with Fiji [15].

## Gene expression analysis using quantitative PCR

C2C12 cells were precultured 6 days with or without 300 ng/mL BMP2 on plastic dish and re-seeded on 12-well plastic plate ($4.2 \times 10^4$ cells/cm$^2$) and cultured with or without BMP2. After 3 days, total RNA was isolated using TRIzol LS reagent (10296010, Thermo Fisher) and a Direct-zol RNA Microprep Kit (R2060, Zymo Research), and cDNA was synthesized using ReverTra Ace qPCR Master Mix with gDNA Remover (FSQ-301, Toyobo). Total RNA was also isolated from MC3T3-E1 cells cultured on a plastic plate, and cDNA was synthesized. Quantitative PCR analysis of genes of interest was conducted on a Viia7 real-time PCR system (Thermo Fisher) with Thunderbird Next SYBR qPCR mix (QPX-201, Toyobo). Oligo Primers (Fasmac) used for analysis are shown in Appendix S2. The standard curve was generated with serial dilution of pooled samples. Expression levels were normalized with *Rpl13a* (60S ribosomal subunit protein gene) as an internal standard [16].

## Fluorescence imaging

For phalloidin staining of cells cultured on micropatterns, cells were fixed 10 minutes with 4% PFA in PBS at RT, washed three times in PBS (5 minutes each) and permeabilized 5 minutes with 0.5% Triton X-100 in PBS. Cells were then washed three times with PBS and incubated 1 hour at RT in staining solution containing Alexa Fluor Plus 647-conjugated phalloidin (1:500 dilution, A30107, Thermo Fisher) and 4′,6-diamidino-2-phenylindole (DAPI, D9542, 2 µg/mL, Sigma-Aldrich). Cells were then washed three more times with PBS (5 minutes each) to remove unbound stains and mounted using ProLong Glass Antifade Mountant (P36980, Thermo Fisher). Fluorescence imaging was performed using Thunder Imaging Systems (Leica).

## Image processing and analysis of dominant orientation

Images were first adjusted to a suitable size based on the "F" mark in the center of the tape for the following analysis. In primary cropping, each cell-confluent rectangle sized 900 µm × 600 µm was picked up, and all vertical rectangles were rotated 90 degrees clockwise. In secondary cropping, each 900 µm × 600 µm rectangle was cropped to 400 µm × 200 µm

with the center of the cell-attached area detected by fluorescence intensity of phalloidin relative to background. Cell-attached areas <80% of rectangle size (600 μm × 300 μm) were excluded. The dominant orientation was analyzed with the OrientationJ plugin in Fiji [17]. All cropping processes were performed using custom-written Macro scripts in Fiji.

### Statistical analysis

Statistical comparison of two independent data groups was performed using Student's t-test. Outliers were determined using Tukey's inner fences. Multiple comparisons, namely ANOVA and Tukey's honestly significant difference (HSD) test, were conducted using IgorPro 9.0 software (HULINKS Inc.).

Descriptive circular statistics and hypothesis testing were performed with the Python package pycircstat2 (0.1.12) and the R circular package (4.3.1) for circular statistics. Half-circle data (−90...+90°) was temporarily converted to full-circle data (0...+360°) before statistical analysis, and angle data was then divided by 2 to return the value to half-circle data. Raw data used for this study were available in Appendix S4.

## Results

### Design and characteristics of customized micropattern tape

Stamping is commonly used to fabricate micropatterns for analysis of cell behavior within confined geometries. However, this method requires a high level of manual skill and precise control over pressure exerted on the surface, which otherwise results in micropattern collapse and limits its reproducibility across users. To simplify this process and increase throughput, we developed customized circular micropatterning tape to enable large-scale, easily accessible fabrication of micropatterns to detect cell chirality (Fig 1A). The design of our micropattern tape incorporated 453 rectangular cutouts arranged in a circle of 19 mm diameter, allowing data acquisition from a large number of rectangles (Fig 1B). The dimensions of individual rectangles (600 μm x 300 μm) were reported in a previous study [18]. Rectangles were arranged in both horizontal and vertical orientations to minimize artifacts potentially resulting from surface inclination or microscopy-dependent shadow effects. Also, to mitigate risk of unexpected flipping of images that might lead to erroneous results, we added an "F" mark at the center of the tape for easy verification of front versus back views (Fig 1D). Furthermore, an extended hemispheric area was incorporated around the micropattern to facilitate handling during tape attachment and removal, thereby preserving pattern integrity during experimental manipulations.

### Micropattern fabrication and cell adhesion on fibronectin islands

We established a standardized fabrication protocol for micropatterns using customized micropattern tape (Fig 2, i). Micropattern tape was adhered to the bottom of a film-based dish, rather than plastic, to facilitate fluorescence imaging (Fig 2, ii). Surface modifications were performed through sequential plasma treatment of the tape to activate the surface and improve adhesive properties inside the micropattern (Fig 2, iii). A 95% ethanol was then added to enhance hydrophilicity, followed by rinsing in RO water (Fig 2, iv). The surface was then coated with a fibronectin solution (40 μg/mL) overnight (Fig 2, v), resulting in formation of discrete fibronectin islands upon tape removal (Fig 2, vi, vii).

To prevent cell adhesion to uncoated areas, a solution of 0.5% Pluronic F-127 was then applied and then removed (Fig 2, viii, ix). After UV sterilization, the micropattern surface was ready for cell seeding (Fig 2, x). Cell suspensions were then added to the prepared surface (Fig 2, xi), and the medium was refreshed once cells had adhered (Fig 2, xii). Over time, multicellular alignment (see Figs 3, 4) became increasingly evident as cells reached confluency, which occurred after several days of culture. Cells were then fixed in 4% PFA/PBS, and actin filaments were stained with phalloidin to visualize cellular alignment.

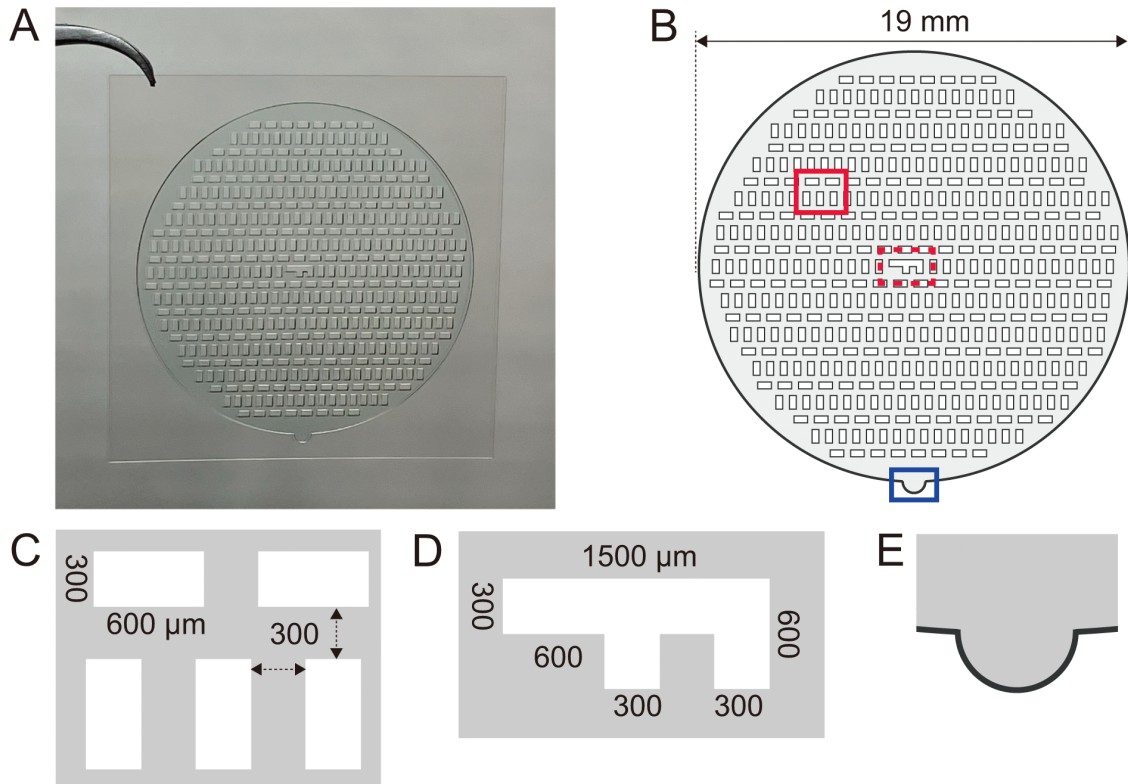

**Fig 1. Design of custom micropattern tape. (A)** Overview of customized tape. **(B)** Characteristic features of micropattern tape. Arrangement of rectangle patterns (red box), "F" mark (dashed red box), and a hemispheric area (blue box). **(C)** Enlarged view of the solid red box in **(B)**. Each rectangle pattern size is 600 μm x 300 μm, and the distance between rectangles in both vertical and horizontal directions is 300 μm (dotted arrows). **(D)** "F" mark in the middle of the micropattern tape (dashed red box in **B**). **(E)** An extended hemispheric area for easy removal of micropattern tape from the film's bottom surface.

## Semi-automated image analysis of cell chirality on the micropattern

A phalloidin image of all micropatterns was obtained using a fluorescence microscope with multipoint Z-focus control and automated stitching. Once the angle of the image was manually adjusted, the image was resized and adjusted for semi-automated workflow using pixel coordinates of the left-top corner of the "F" mark as a positional reference. To quantitatively analyze cell alignment on micropattern rectangles, we developed semi-automated image analysis in Fiji [15] comprised of five steps: primary cropping, rotation, secondary cropping, measurement of dominant orientation, and analysis of frequency distribution (Fig 3A). In brief, all rectangles were individually cropped (Fig 3A,1), and vertically-oriented rectangles were rotated clockwise to align horizontally (Fig 3A,2). Next, we conducted secondary cropping, excluding border cells aligned along edges of rectangles (Fig 3A,3). The dominant orientation of actin filaments within central regions was then quantified using the OrientationJ plugin in Fiji (Fig 3A,4). This plugin employs structural tensor analysis by calculating pixel intensity gradients in different directions to characterize orientation and to quantify dominant alignment angles (θ) for the cropped central area of each rectangle. Finally, dominant orientation data from all rectangles was compiled to generate a frequency distribution of orientation, which indicated cell chirality (Fig 3A,5). This workflow allowed high-throughput quantification of cell alignment.

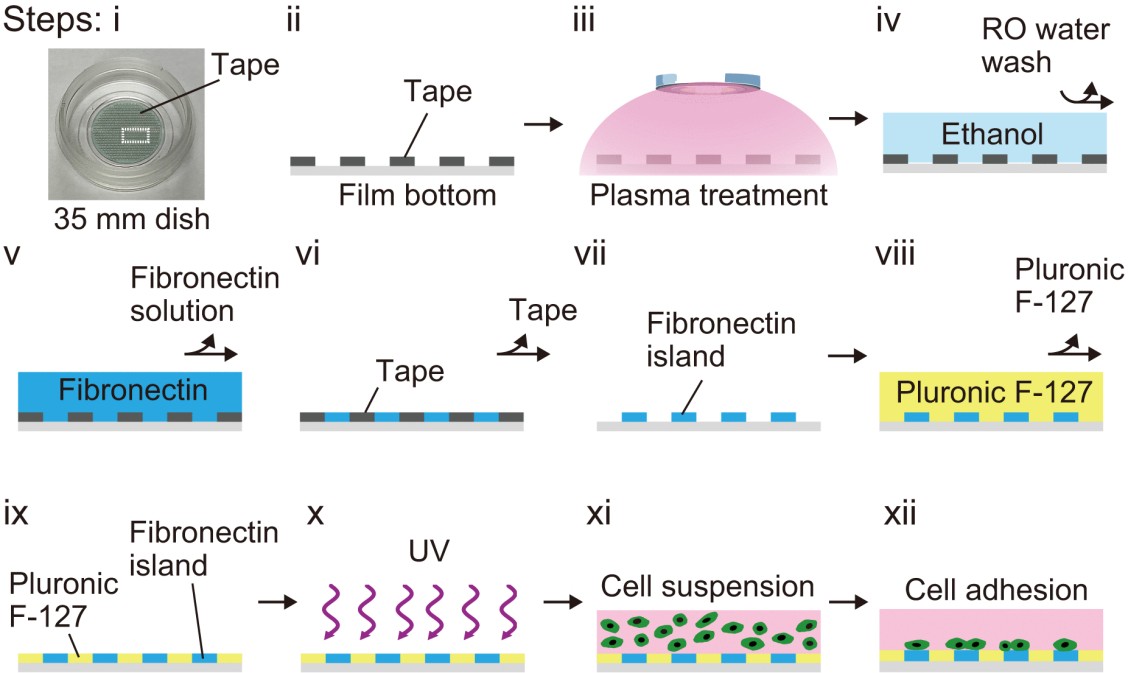

**Fig 2. Protocol for cell culture on micropatterns.** (i) Image shows tape attached to the film bottom of a 35 mm dish. (ii-ix) Schematics of steps, starting with a side view (ii) of area boxed in (i). After plasma treatment (iii), the surface was treated with ethanol to reduce interfacial tension (iv) and washed with water. After fibronectin coating overnight (v), the fibronectin solution was aspirated (vi) and the tape removed (vii). Pluronic F-127 solution was added to prevent cell adhesion to uncoated areas (viii) followed by washing with water (ix). Following UV treatment (x), cell suspensions were added (xi), and the culture medium was changed after cells adhered (xii).

### Definition of cell chirality and validation of the micropattern tape method

To validate micropattern methodology, we assessed chirality of C2C12 cells, a myoblastic line previously reported to exhibit asymmetric alignment [13]. To do so, C2C12 cells were cultured on micropatterned rectangles, stained with phalloidin for actin filaments, and imaged by microscopy (Fig 3B). Using the semi-automated workflow described above, each micropattern rectangle of 600 µm × 300 µm (Fig 3C) was further cropped to isolate the central 400 × 200 µm region (Fig 3D). Orientations of actin filaments were visualized by color-coded orientation (Fig 3E) relative to the horizontal axis using the angle color map (Fig 3F). The dominant orientation (θ) of each central region was calculated using the OrientationJ plugin and served as a means to identify cell chirality (Fig 3E). In this study, the horizontal axis was set parallel to the long axis of the rectangle (θ = 0°). Cells aligned in the left-up to the right-down direction (θ < 0°) were classified as clockwise (CW), whereas cells aligned in the left-down to right-up direction (θ > 0°) were classified as counterclockwise (CCW) (Fig 3G). The circular mean was calculated from the dominant orientations to define chirality of cell populations. Based on these parameters, C2C12 cells exhibit CW chirality (Fig 3H).

To validate reproducibility of the cell chirality assay, we repeated C2C12 cell experiments and compared them with analysis of the mouse osteoblast cell line MC3T3-E1, which exhibits chirality opposite to C2C12 cells [13]. Both cell types were cultured on the micropattern surface (Fig 4A), and the central 400 × 200 µm regions of each rectangle were cropped (Fig 4B). The dominant orientation of actin filaments in central regions was quantified using OrientationJ (Fig 4C). Distribution analysis of dominant orientations showed that C2C12 cells aligned predominantly in a CW direction, whereas MC3T3-E1 cells exhibited a CCW direction (Fig 4D). These results indicate that both cell lines show expected opposite

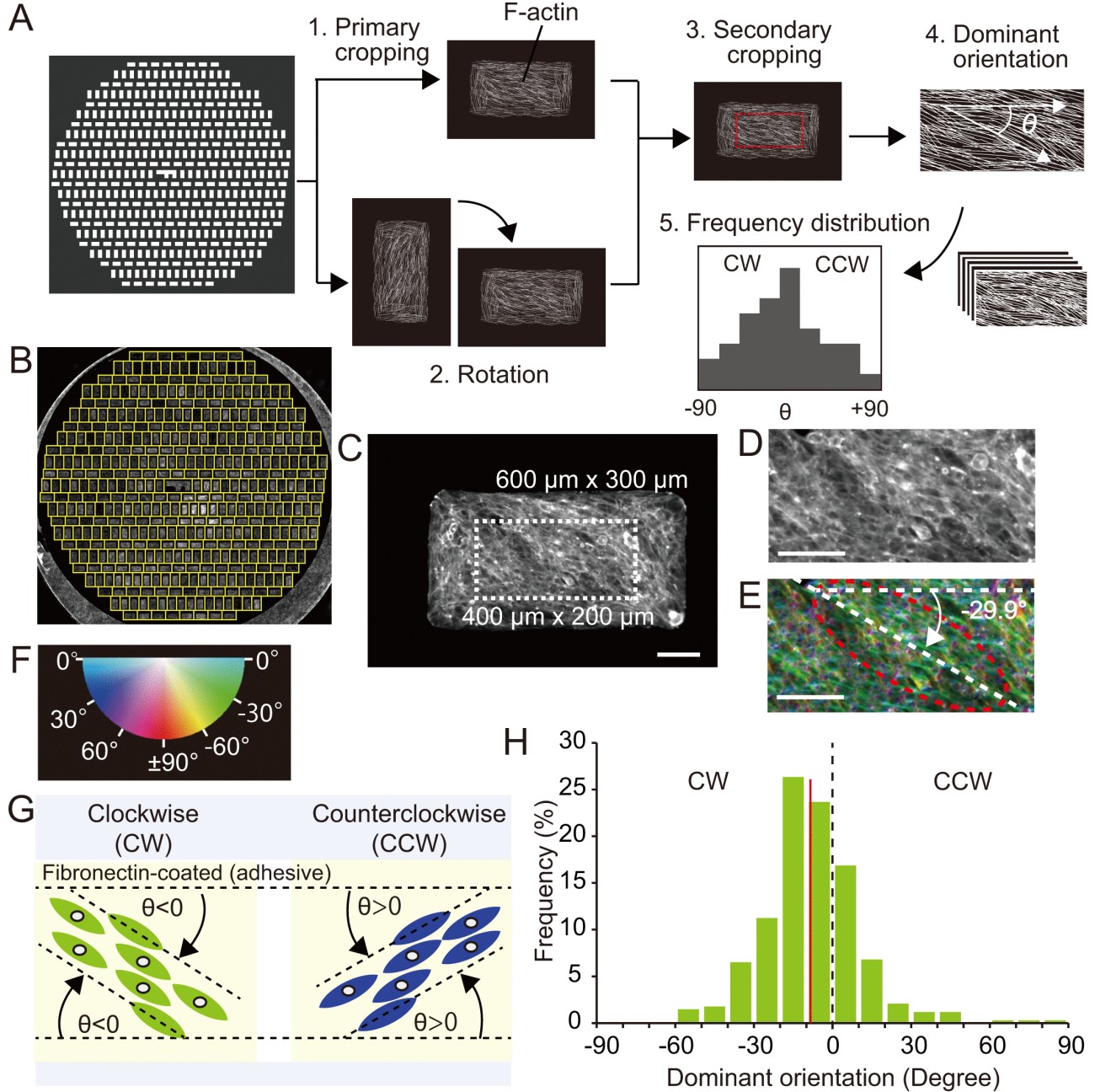

**Fig 3. Quantitative cell chirality analysis based on dominant orientation on micropatterns. (A)** Semi-automated analysis of dominant orientation. 1) Primary cropping of each micropattern from fluorescence images of phalloidin-stained cells. 2) Rotation of vertical micropattern to horizontal. 3) Secondary cropping of the central area (red box). 4) Dominant orientation analysis using OrientationJ, 5) Frequency distribution of the dominant orientation. **(B)** Each rectangle was cropped in the primary cropping. Scale bar, 1 mm. **(C)** The central area (white dotted area) was cropped in secondary cropping. Central cropping size, 400 μm x 200 μm. Scale bar, 100 μm. **(D)** The cropped area from **(C)**. Scale bar, 100 μm. **(E)** Calculation of the dominant orientation. The long axis of the red dotted ellipse shows the dominant orientation. Eccentricity of ellipses demonstrates calculated coherence of orientation. White dashed lines represent the angle of the dominant orientation of the cropped area. **(F)** Color map of orientation. **(G)** Schematic depiction of cells from the culture medium side. Left-up to right-down orientation was defined as clockwise (CW) alignment, and left-down to right-up orientation as counterclockwise (CCW). **(H)** Frequency distribution of the dominant orientation on a micropattern. The circular mean (red line) = −8.52°; circular standard deviation (SD) = 17.82°; n = 338. Testing for circular mean angle by a one-sample test. Since a hypothetical population mean ($\mu0$) = 0 lies outside the 95% confidence interval (CI) of the mean for the population ($\mu$) [−10.65° to −6.38°], the null hypothesis ($\mu0 = \mu$) is rejected.

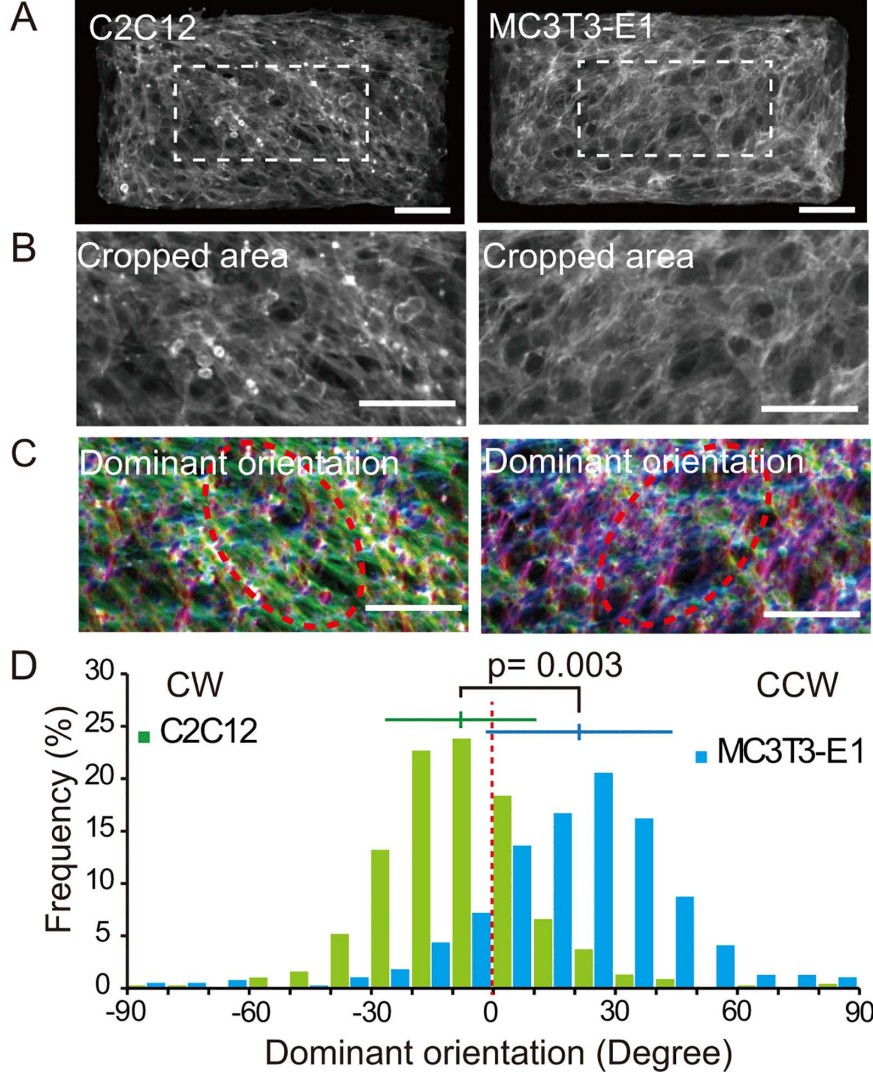

**Fig 4. C2C12 and MC3T3-E1 cells show opposite chirality on the micropattern. (A)** C2C12 and MC3T3-E1 cells on the micropattern. **(B)** Cropped areas used to assess the dominant orientation. **(C)** Orientation analysis. See Fig 3F for color map. Scale bars, 100 μm. **(D)** Frequency of the dominant orientation of C2C12 (green) and MC3T3-E1 (blue) cells. C2C12 cells: circular mean = −7.62°, circular SD = 18.45°, n = 697. Analysis of the mean angle by a one-sample test. Since μ0 = 0 (red dotted line) lies outside the 95% CI of μ [−9.14° to −6.11°], the null hypothesis (μ0 = μ) is rejected. MC3T3-E1 cells: circular mean = 21.22°, circular SD = 22.70°, n = 389. Analysis of the mean angle by a one-sample test. Since μ0 = 0 lies outside the 95% CI of μ [18.86° to 23.59°], the null hypothesis (μ0 = μ) is rejected. p = 0.003, Watson's $U^2$-test. Horizontal bars indicate circular mean ± SD.

bias in cell alignment, confirming that our micropattern method is a reliable approach to detect and compare cell chirality across different cell types.

### Optimization of chirality assay conditions for C2C12 cells undergoing osteoblastic differentiation induced by BMP2

To determine whether cell chirality depends on cell differentiation states, we optimized culture conditions for BMP2-induced osteoblastic differentiation of C2C12 cells. We first compared two different culture media, αMEM and DMEM, in the absence or presence of BMP2. Osteoblastic differentiation in 6 days was assessed by measuring activity of ALP, a

well-established osteoblast differentiation marker. BMP2-treated C2C12 cells exhibited significantly higher ALP activity when grown in αMEM compared to DMEM, while ALP activity in cells not treated with BMP2 was undetectable in cells grown in either medium (Fig 5 A,5B). These results indicate that αMEM is the optimal medium for analysis of BMP2-induced osteoblastic differentiation. To confirm myoblastic to osteoblastic differentiation of C2C12 cells, we performed Immunofluorescence staining to assess expression of myosin heavy chain (MHC) and runt-related transcription factor 2 (RUNX2), respective markers of myoblasts and osteoblasts. BMP2-treated C2C12 cells exhibited elevated RUNX2 expression, indicative of induced osteoblastic differentiation, and decreased MHC expression, indicating suppression of myoblastic differentiation (Fig 5C). These results confirm that BMP2-treated C2C12 cells show robust osteoblastic differentiation when grown in αMEM.

To determine how different coating substrates might impact osteoblastic differentiation, we compared three commonly utilized extracellular matrix components: fibronectin, vitronectin, and collagen type 1 (Col1). Specifically, we seeded C2C12 cells on culture dishes coated with these substances and evaluated both MHC immunostaining and ALP activity, with or without BMP2 treatment. Among substrates, Col1 showed decreased MHC expression relative to other conditions in the absence of BMP2 (Fig 5D). Following osteoblast differentiation by BMP2, vitronectin coating promoted the most robust ALP activity compared with fibronectin and Col1 (Fig 5E). Notably, fibronectin coating markedly suppressed ALP activity after osteoblastic differentiation (Fig 5E). These findings suggest that fibronectin and Col1 are less suitable coatings than vitronectin for assessment of osteoblastic differentiation.

## C2C12 cells show CCW cell chirality after osteoblastic differentiation

To examine the effect of changing coating substrates from fibronectin to vitronectin on cell chirality, we first cultured C2C12 cells in αMEM on fibronectin- and vitronectin-coated micropatterns in the absence of BMP2 and assessed the distribution of dominant orientation. The height and negative bias of the dominant orientation histogram of C2C12 cells on vitronectin-coated micropatterns were both decreased compared with effects seen on fibronectin-coated micropatterns (kurtosis, 1.97 to 0.14, circular mean −7.48 to −1.10) (Fig 6A). Consistently, the average value of the dominant orientation of MC3T3-E1 cells was smaller on vitronectin than on fibronectin (Appendix S3 A). Since the C2C12 and MC3T3-E1 cells exhibited opposite chirality on both vitronectin and fibronectin, it can be concluded that the two cell types exhibit different chirality regardless of the substrate. Next, to determine the impact of BMP2-induced osteoblastic differentiation on cell chirality, we cultured C2C12 cells on vitronectin-micropatterns in αMEM plus BMP2 for 3 more days after 6-day preculture (total 9 days) in the same differentiation media in plastic dish. Strikingly, the distribution of dominant orientation of C2C12 cells was reversed from CW to CCW after treatment with BMP2 (Fig 6B). After five days of BMP2 stimulation, heterogeneity in dominant orientation of C2C12 cells increased, and some micropatterns began exhibiting CCW chirality (Appendix S3 B). Thus, the dominant orientation gradually shifted from CW to CCW during the transdifferentiation from myoblastic to osteoblastic cells. The consistent results were also observed in the dose-dependency of BMP2 (Appendix S3 C).

To investigate the molecular mechanisms underlying the opposite cell chirality of C2C12 and MC3T3-E1 cells, as well as the shift in chirality of C2C12 cells during BMP2-induced transdifferentiation, we performed a quantitative PCR analysis of several actin cytoskeleton regulators, *Actn1* [5], *Pfn1* [18], *Fscn1* [19], and *Fmn1* [20], that have been reported to contribute to cell chirality determination. We also included *Actn2*, which is expressed in myoblast [21]. As shown in Fig 6C, *Pfn1, Actn1* and *Actn2* expression was lower in MC3T3-E1 cells compared with control C2C12 cells. Following BMP2 stimulation, expression of *Pfn1, Actn1* and *Actn2* was decreased in the direction towards the levels in MC3T3-E1 cells. By contrast, *Fscn1* and *Fmn1* expression was higher in MC3T3-E1 cells than in C2C12 cells. BMP2-induced C2C12 cells did not show increased *Fscn1* and *Fmn1* expression. These data suggest that *Actn1, Actn2* and *Pfn1* expression levels might be associated with CW to CCW shift in C2C12 cells during BMP2-stimulation, and with CCW chirality of MC3T3-E1 cells.

We also analyzed the expression of osteoblastic (*Runx2* and *Alpl*) and myoblastic markers (*Myog, Ckm*, and *Myh1*). Upregulation of *Runx2* and *Alpl*, and downregulation of *Myog, Ckm*, and *Myh1* in BMP2-induced C2C12 cells indicates

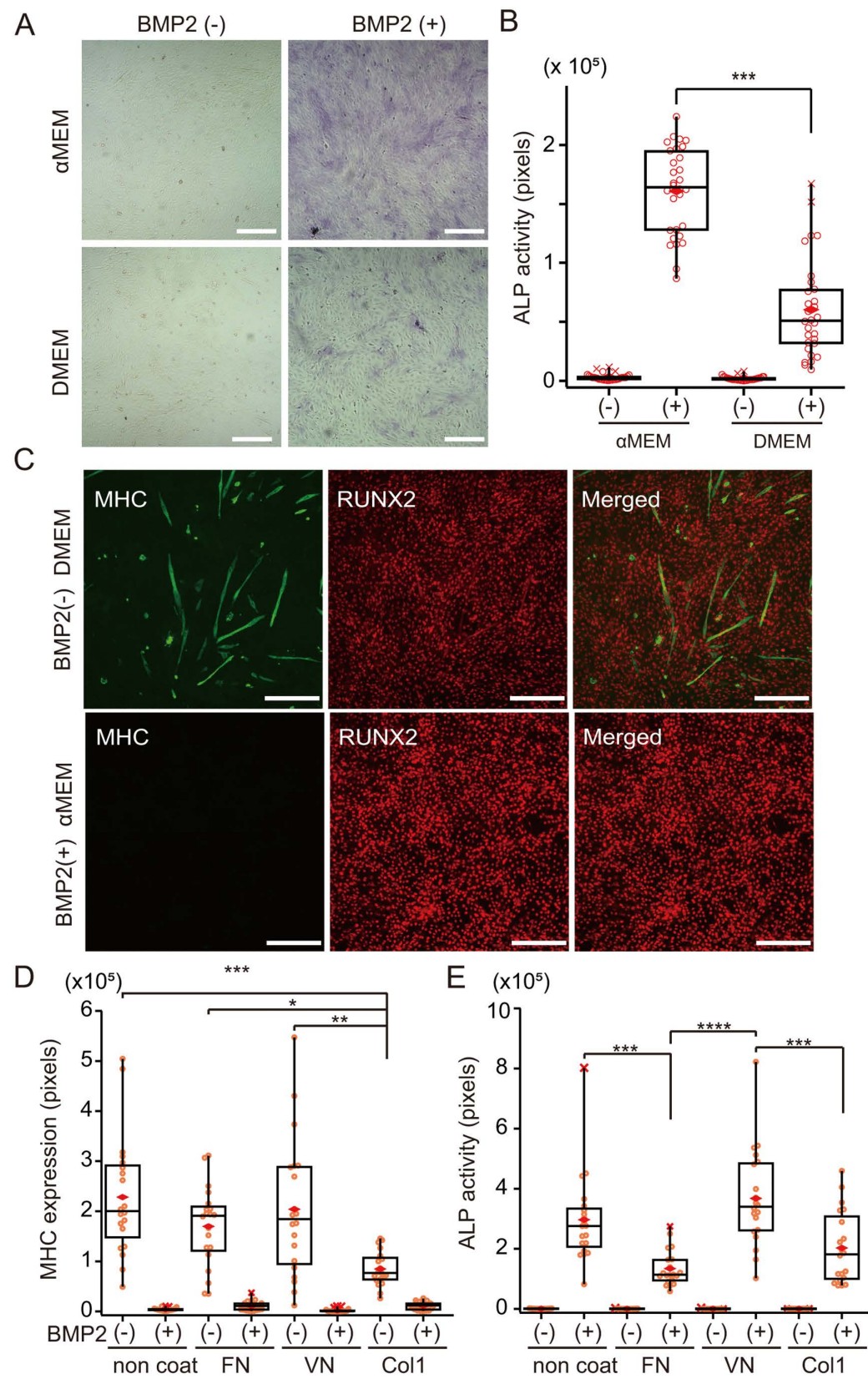

**Fig 5. BMP2-induced osteoblastic differentiation of C2C12 cells. (A)** ALP assay of untreated and BMP2-treated C2C12 cells cultured in αMEM or DMEM medium. Scale bar, 300 μm. **(B)** Quantification of the ALP-activity positive area shown in **(A)**. n = 30 each, 5 areas/well, 6 wells. ALP-positive areas were quantified in 5 random ROIs (1647 μm x 1647 μm, pixel area: 512 x 512) per dish for each condition and were plotted as open circles. X, outlier. ***p < 0.001, Student's t-test. Box plots show the average, median, and quartiles. **(C)** Immunofluorescence staining of the myogenic MHC marker and the osteogenic RUNX2 marker. Scale bar, 300 μm. **(D)** Quantification of MHC-positive areas on various coating substances. FN, fibronectin. VN, vitronectin. Col1, type 1 collagen. n = 20 for each group. **(E)** Quantification of ALP activity-positive areas on various coating substances. n = 20 for each group. Statistical significance, *p < 0.05, **p < 0.01, ***p < 0.001 and ****p < 0.0001, Tukey's HSD test.

transdifferentiation from myoblastic to osteoblastic cells by BMP2 stimulation. Downregulation in *Actn2* expression is also consistent with the loss of myoblast markers since alpha-actinin-2 encoded by *Actn2* anchors actin filaments to the muscle Z-line [21]. These data suggest that cell chirality is not fixed but is rather a dynamic, environmentally-responsive property that can be altered during differentiation in response to osteogenic differentiation cues.

## Discussion

In this study, we developed a micropattern tape method that simplifies fabrication of cell adhesive coating patterns, allowing high-throughput analysis of cell chirality. Although here we used our method to assess myoblast and osteoblast chirality, the method can be adapted to other cell types and used to assess variation of cell chirality under different conditions, including cellular differentiation and drug screening.

To successfully implement this method, several parameters require careful optimization. Key parameters such as cell seeding density significantly influenced the method's efficacy: we found that insufficient cell density resulted in sparse adhesion and poor micropattern formation, while overly high density caused cell overcrowding, which can obscure cellular alignment (Appendix S1 A-C). In our protocol, we gently moved dishes back-and-forth and side-to-side five times periodically before cells could adhere to micropatterns, to prevent cells from concentrating in the center of the dish and ensure their even distribution. The smaller the initial or final cell numbers on individual micropatterns, the greater the angular variability and the smaller the kurtosis (Appendix S1 D-E). Therefore, the even distribution of cells within the dish is important to ensure a sufficient number of cells on the micropattern for determination of collective cell chirality. We also found that ethanol treatment during micropattern preparation must not exceed 10 seconds to prevent tape damage. Proper adhesion of the tape to the dish surface was ensured by tightly pressing the tape to prevent adhesion of coating proteins (fibronectin or vitronectin) to undesired areas. Moreover, selection of surface materials, such as plastic, glass, and film-based, for the culture dish is a critical factor influencing the method's effectiveness. We selected film-bottomed surfaces over glass as the former decreased unintended cell adhesion outside of micropatterns after tape removal. Even when treated with anti-adhesive reagents like Pluronic F-127, we observed glass surfaces still promoted unintended cell attachment outside the micropattern, disrupting alignment within patterned areas and compromising results. By contrast, we found that film-bottomed surfaces reduced adhesion of cells outside patterned areas and helped maintain micropattern integrity.

In this study, we used fluorescence imaging to obtain an overall view of micropatterns. However, careful considerations are required when using phase contrast microscopy to image cells on micropatterns, as potential shadow effects can cause artifacts and distort imaging results. To mitigate this effect, we incorporated a mix of vertical and horizontal rectangles in our micropattern design (Fig 1). In the presence of shadow effects, directional artifacts due to oblique illumination, dominant orientations derived from vertical and horizontal rectangles display distinct dominant orientations, while in the absence of shadow effects, these two groups should display identical dominant orientations. Another common issue encountered during imaging is unintentional image inversion, which can occur at specific microscope settings and, if unnoticed, lead to misinterpretation of cell chirality. Here, the orientation of the "F" mark functioned as simple and effective confirmation of correct imaging orientation, front or back, prior to data analysis. Overall, these considerations ensure reliability and reproducibility of cell chirality analysis using this method.

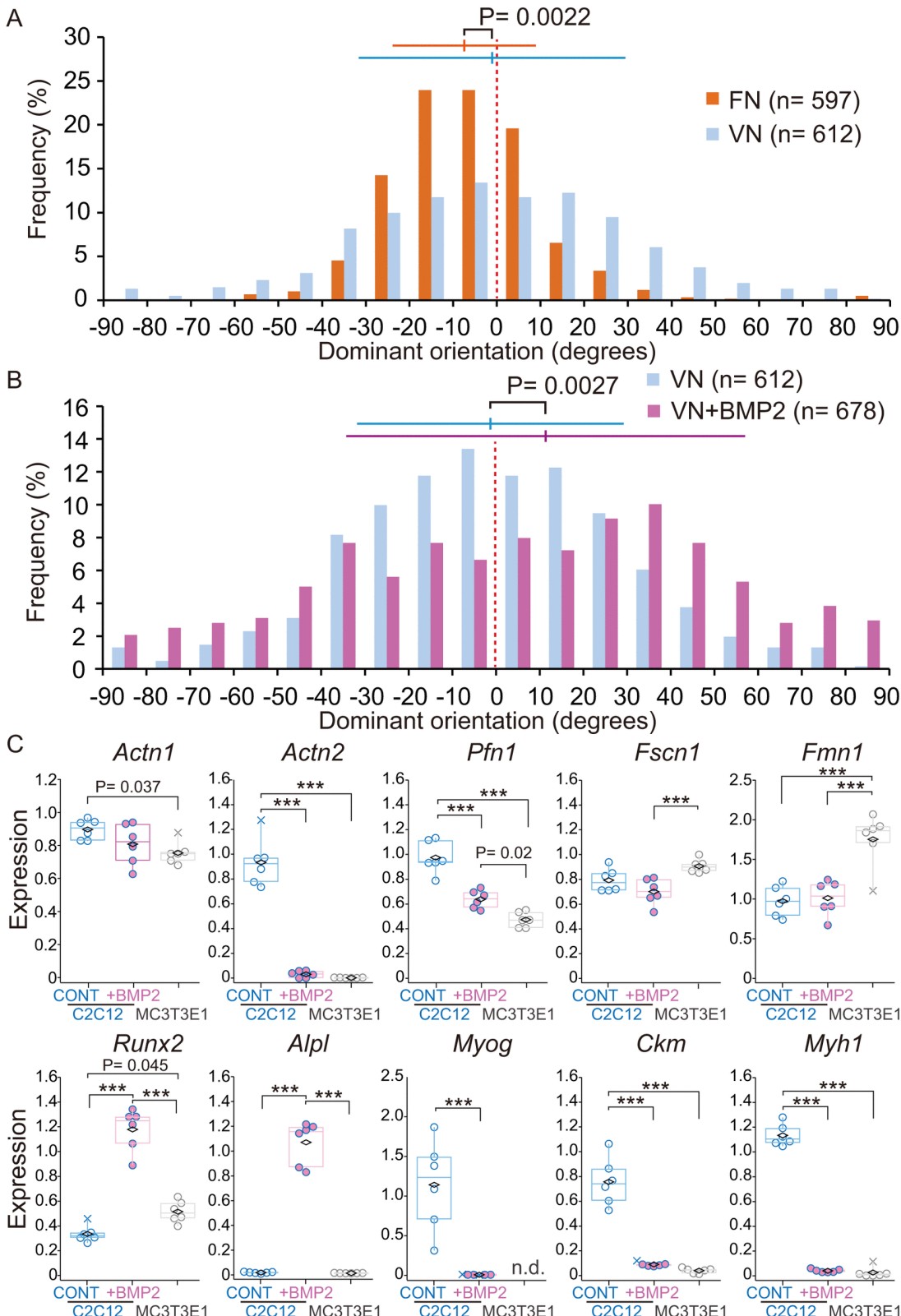

**Fig 6. BMP2-induced changes in chirality and gene expression of C2C12 cells. (A)** Frequency of dominant orientation of C2C12 cells grown on fibronectin-coated micropatterns (FN, circular mean = −7.48°, circular SD = 16.40°, n = 597) and vitronectin-coated micropatterns (VN, circular

mean = −1.10°, circular SD = 30.57°, n = 612, also shown in **(B)**), p = 0.0022, Watson's U²-test. **(B)** Frequency of the dominant orientation of BMP2-treated C2C12 cells grown on vitronectin-coated micropatterns (VN + BMP2, circular mean = 11.56°, circular SD = 45.65°, n = 678), p = 0.0027, Watson's U²-test. Horizontal bars indicate circular mean ± SD. **(C)** Gene expression of actin cytoskeleton regulators, associated with cellular chirality, and osteoblastic or myoblastic markers. Relative expression levels were normalized with *rpl13a* (60S ribosomal subunit protein gene) as an internal standard. CONT, C2C12 cells cultured without BMP2. Each data is plotted as open circles. X, outlier, diamond, mean; N = 6. *** P < 0.001. ANOVA, Tukey HSD.

There are several other methods for creating micropatterns, including PDMS stamping [5,22] and stencil methods [18]. The PDMS stamping method requires the creation of a mold. Molds can be made using cutting processes [23], 3D printing [24], or using photoresist [25]. The tape method has several advantages. It allows for the easy, stable fabrication of a large number of well-shaped micropatterns with a high yield. The tape itself is also simple to remove as opposed to a hardened resin mask outside the micropattern in the stencil method [18]. However, there are two limitations to this method. First, there is a size limitation. Designs of the micropattern with a scale of a few micrometers are too small to be excised out. Second, the inner enclosed part within a micropattern, such as a circle within a ring, is difficult to make. The inner part falls out during the cutting process of the tape.

This tape method enables a larger sample size per experiment, leading to a more robust and quantitative evaluation. In our hands, methods such as PDMS stamping were less stable, and required tedious exclusion of micropatterns with poor morphology. However, the tape method produces micropatterns with consistent, stable shapes, which provides at least comparable accuracy regardless of the amount of technical experience. Furthermore, the analysis system automates the process of cropping and preparing numerous micropatterns for analysis. Its accuracy depends on the use of the widely adopted ImageJ plugin, OrientationJ, for angle calculation [18,26].

Based on our observations in myoblasts and osteoblasts, cell chirality is likely cell type-specific. We validated our method by analyzing chirality on micropatterns of C2C12 myoblasts and MC3T3-E1 osteoblasts (Fig 4D). C2C12 myoblasts displayed stable CW cell chirality, while MC3T3-E1 osteoblasts showed the opposite CCW cell chirality. Moreover, we observed that BMP2 triggered a positive shift in distribution of the dominant orientation from myoblastic (CW) to osteoblastic (CCW) chirality of C2C12 cells on vitronectin-coated micropatterns, suggesting that cell chirality may change during lineage specification in response to differentiation-associated cytoskeletal or signaling changes. We performed a gene expression analysis of several actin cytoskeleton regulators: *Actn1* [5], *Actn2* [21], *Pfn1* [18], *Fscn1* [19], and *Fmn1* [20]. The expression levels of *Actn1, Actn2* and *Pfn1* might account for the dominant orientation differences between C2C12 and MC3T3-E1 cells, as well as with the chirality shift induced by BMP2 (Fig 6). Previous study demonstrated that *Pfn1* knockdown in human fibroblast shifts chirality from CW to CCW [18], suggesting that observed downregulation of *Pfn1* in BMP2-induced C2C12 cell may indeed causatively shift the cell chirality from CW to CCW. In other words, and that lower *Pfn1* expression in the presence of Actn1 favors CCW chirality [18]. Interestingly, others have reported a chirality shift from CCW to CW by differentiating mesenchymal stem cells to adipocytes [27], in contrast to CW to CCW shift from myoblasts to osteoblasts (Fig 6B). Changes in cell chirality may also occur in cell aging considering a reported association between aging and cytoskeleton organization in mesenchymal stem cells under self-renewal and osteogenic conditions [28]. Curiously, the chirality of an extracellular matrix may affect differentiation of mesenchymal stem cells to osteogenic and adipogenic cells possibly through cell chirality modulation [22].

The tape method with high density micropatterns reported here provides a versatile analytical tool allowing quantitative measurement of the cell chirality spectrum in terms of distribution of the dominant orientation. Even cells with the same handedness showed significant shifts in distribution of the dominant orientation (Fig 6A). Although established cell lines were used in this study, the tape method also facilitates chirality analysis of primary cells. Combining our tape method with in vivo chirality analysis during development [29,30] could reveal the biological significance of cell chirality in vivo. This study and previous reports collectively indicate that cell chirality is a dynamic and stimulus-responsive property reflecting cytoskeletal remodeling.

## Supporting information

**S1 Appendix. Optimization of cell seeding density for tape method.**
(DOCX)

**S2 Appendix. List of primers for gene expression analysis.**
(PDF)

**S3 Appendix. Effect of coating substrates and conditions of BMP2 treatment on the dominant orientation.**
(DOCX)

**S4 Appendix. Raw data used for this study.**
(XLSX)

## Acknowledgments

We thank Rei Kuwabara (Keio Global Science Campus) for discussion, the staff of the Core Facility for their technical assistance and support throughout this study, and Elise Lamar for English editing.

## Author contributions

**Conceptualization:** Qingkai Weng, Yukiko Kuroda, Koichi Matsuo, Katsuhiro Kawaai.

**Data curation:** Qingkai Weng, Koichi Matsuo, Katsuhiro Kawaai.

**Formal analysis:** Qingkai Weng, Koichi Matsuo, Katsuhiro Kawaai.

**Funding acquisition:** Koichi Matsuo.

**Investigation:** Qingkai Weng, Katsuhiro Kawaai.

**Methodology:** Takashi Osaka, Hiroaki Onoe, Koki Yoshida.

**Project administration:** Koichi Matsuo, Katsuhiro Kawaai.

**Software:** Qingkai Weng, Koichi Matsuo.

**Supervision:** Yukiko Kuroda, Koichi Matsuo, Katsuhiro Kawaai.

**Visualization:** Qingkai Weng, Koichi Matsuo, Katsuhiro Kawaai.

**Writing – original draft:** Qingkai Weng, Koichi Matsuo, Katsuhiro Kawaai.

**Writing – review & editing:** Takashi Osaka, Hiroaki Onoe, Koki Yoshida, Yukiko Kuroda, Koichi Matsuo, Katsuhiro Kawaai.

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
