## [Decision Letter · Decision Letter 0]

14 Aug 2025

Dear Dr. Kawaai,

Thank you for submitting your manuscript to PLOS ONE. After careful consideration, we feel that it has merit but does not fully meet PLOS ONE’s publication criteria as it currently stands. Therefore, we invite you to submit a revised version of the manuscript that addresses the points raised during the review process.

We look forward to receiving your revised manuscript.

Kind regards,

Baeckkyoung Sung, Ph.D.

Academic Editor

PLOS ONE

Journal Requirements:

This research was supported by the JSPS KAKENHI grant number 21H05789, 21H03060 and 23K21467

We thank financial support from the Keio Global Science Campus, and the staff of the Core Facility for their technical assistance and support throughout this study and Elise Lamar for English editing. This work was supported by the Japan Society for the Promotion of Science [grant numbers 21H05789, 21H03060 and 23K21467].

This research was supported by the JSPS KAKENHI grant number 21H05789, 21H03060 and 23K21467

NO authors have competing interests

Additional Editor Comments:

The authors are suggested to revise the manuscript in line with the reviewers' comments.

Reviewers' comments:

Reviewer's Responses to Questions

**Comments to the Author**

1. Is the manuscript technically sound, and do the data support the conclusions?

Reviewer #1: Yes

Reviewer #2: Yes

2. Has the statistical analysis been performed appropriately and rigorously?

Reviewer #1: Yes

Reviewer #2: Yes

3. Have the authors made all data underlying the findings in their manuscript fully available?

Reviewer #1: Yes

Reviewer #2: Yes

4. Is the manuscript presented in an intelligible fashion and written in standard English?

Reviewer #1: Yes

Reviewer #2: Yes

Reviewer #1: This study presents a micropattern tape method for fabricating micropatterns to analyze cell chirality. While the technique demonstrates potential for detecting chirality and reveals plasticity during differentiation, the results appear relatively simplistic, with limited parametric investigation and minimal theoretical analysis.

Concerns are listed below:

1. The reported chirality reversal in C2C12 cells following BMP2-induced differentiation lacks intuitive evidence. More comprehensive dose-response and time-course studies are needed to clarify the transformation dynamics and identify key influencing factors.

2. The manuscript primarily reports observations without sufficient mechanistic or theoretical discussion. For instance, what underlying biological or physical principles account for the opposite chirality exhibited by C2C12 versus MC3T3-E1 cells? A deeper analysis of these findings would significantly strengthen the work.

3. The optimization of cell density for even distribution is noted, but the rationale remains unclear. Is monolayer formation essential for proper observation?

4. The choice of C2C12 and MC3T3-E1 cells needs stronger biological rationale beyond their chirality differences. What controls were implemented to isolate chirality effects from other substrate/environmental influences? Would inclusion of additional cell types help validate method generalizability?

5. The manuscript contains inconsistent capitalization of figure panel labels (e.g., uppercase A/B/C vs. lowercase a/b/c).

Reviewer #2: Manuscript: PONE-D-25-25096

Title: Use of a New Micropattern Tape Method to Detect Chirality shifts in differentiating C2C12 Cells

General Comments: Chirality is a property of cells demonstrating right and left behavior in terms of morphology and organization. The present paper develops a new method to pattern small regions of cells applied in rectangular micro patterns for chirality analysis. The authors develop a unique method utilizing a tape system which enables application of a range of extracellular matrix proteins while creating gap regions between the applied cell areas. They also have developed a sophisticated optical microscopic analysis system to determine orientation and a processing method for quantitation of images obtained. Using this method they look at the impact of an exogenous agent - bone morphogenic protein on alteration of chirality of two cell lines C2C 12 and MC3T3E1. They specifically show that the system can detect plasticity of cell chirality in vitro. The paper in general is well written is clear and describes the unique method. There are few areas which may be addressed to strengthen the paper

General issues to be discussed:

1. Provide a bit more in the discussion of other existing in vitro fabrication methods that other investigators are using and provide a bit more point by point detail of why your method is better.

2. Please add detail as to the nature of the tape, its constituents, its composition and the nature of the adhesive.

3. Please add detail as to the uncoated film bottom dish. What is this material? Is this available for other investigators. In general, both of these details in points #2 and #3 above are important if a method is going to be more universally applied by others.

4. For figure 5 where there is reference to a range of extracellular matrix proteins please clarify if this involved the same general coating technique of the dish/microislands or is this different?? This is confusing to the reader.

5. In the discussion at the beginning in the first sentence it states, “cell adhesive fibronectin patterns”. Related to point #4 if this is only for fibronectin that is fine. However, if other adhesive proteins are being utilized here as well this is confusing to the reader and needs to be clarified.

6. On page 18 lines 390 through 392 need to be clarified. Specifically, the issue of what is being adhered to what, and to prevent adhesion to uncoated areas is unclear to the reader - please explain.

7. In the manuscript the word “holes” was used to designate the gap areas. Would use a different term such as uncoated area or region. For most readers holes will convey a mental image of something round rather than the rectangular regions you actually create.

8. would provide a bit more in the discussion re the sensitivity of this techniques and the optical/image analysis system you developed. How sensitive is this? How accurate? How does this compare to other systems.

9. As to the biological findings of chirality changing with differentiation and cell aging or ECM effects – would expand on this a bit with discussions as to what is known in the literature and how do your findings conform or extend the body of knowledge.

**Do you want your identity to be public for this peer review?** For information about this choice, including consent withdrawal, please see our Privacy Policy

Reviewer #1: No

Reviewer #2: **Yes: ** Marvin J Slepian

---

## [Author Response · Author response to Decision Letter 1]

7 Oct 2025

PONE-D-25-25096

Use of a new micropattern tape method to detect chirality shifts in differentiating C2C12 cells

Point-by-point response to the comments

Reviewer #1: This study presents a micropattern tape method for fabricating micropatterns to analyze cell chirality. While the technique demonstrates potential for detecting chirality and reveals plasticity during differentiation, the results appear relatively simplistic, with limited parametric investigation and minimal theoretical analysis.

Concerns are listed below:

1. The reported chirality reversal in C2C12 cells following BMP2-induced differentiation lacks intuitive evidence. More comprehensive dose-response and time-course studies are needed to clarify the transformation dynamics and identify key influencing factors.

Following the reviewer's suggestion, we performed additional experiments and presented the time course (Appendix S3B) and dose dependency (Appendix S3C) of the change in dominant orientation induced by BMP2. We explained these new data in the main text in the revised manuscript as follows:

In results:

”After five days of BMP2 stimulation, heterogeneity in dominant orientation of C2C12 cells increased, and some micropatterns began exhibiting CCW chirality (Appendix S3B). Thus, the dominant orientation gradually shifted from CW to CCW during the transdifferentiation from myoblastic to osteoblastic cells. The consistent results were also observed in the dose-dependency of BMP2 (Appendix S3C).“

2. The manuscript primarily reports observations without sufficient mechanistic or theoretical discussion. For instance, what underlying biological or physical principles account for the opposite chirality exhibited by C2C12 versus MC3T3-E1 cells? A deeper analysis of these findings would significantly strengthen the work.

To address the reviewer's comments, we have performed additional experiments and added the following text to the results and discussion section of the revised manuscript.

In results:

“To investigate the molecular mechanisms underlying the opposite cell chirality of C2C12 and MC3T3-E1 cells, as well as the shift in chirality of C2C12 cells during BMP2-induced transdifferentiation, we performed a quantitative PCR analysis of several actin cytoskeleton regulators, Actn1 [5], Pfn1 [18], Fscn1 [19], and Fmn1 [20], that have been reported to contribute to cell chirality determination. We also included Actn2, which is expressed in myoblast [21]. As shown in Fig. 6C, Pfn1, Actn1 and Actn2 expression was lower in MC3T3-E1 cells compared with control C2C12 cells. Following BMP2 stimulation, expression of Pfn1, Actn1 and Actn2 was decreased in the direction towards the levels in MC3T3-E1 cells. By contrast, Fscn1 and Fmn1 expression was higher in MC3T3-E1 cells than in C2C12 cells. BMP2-induced C2C12 cells did not show increased Fscn1 and Fmn1 expression. These data suggest that Actn1, Actn2 and Pfn1 expression levels might be associated with CW to CCW shift in C2C12 cells during BMP2-stimulation, and with CCW chirality of MC3T3-E1 cells.

We also analyzed the expression of osteoblastic (Runx2 and Alpl) and myoblastic markers (Myog, Ckm, and Myh1). Upregulation of Runx2 and Alpl, and downregulation of Myog, Ckm, and Myh1 in BMP2-induced C2C12 cells indicates transdifferentiation from myoblastic to osteoblastic cells by BMP2 stimulation. Downregulation in Actn2 expression is also consistent with the loss of myoblast markers since alpha-actining-2 encoded by Actn2 anchors actin filaments to the muscle Z-line [21].”

In discussion:

We performed a gene expression analysis of several actin cytoskeleton regulators: Actn1[5], Actn2 [21], Pfn1[18], Fscn1[19], and Fmn1[20]. The expression levels of Actn1, Actn2 and Pfn1 might account for the dominant orientation differences between C2C12 and MC3T3-E1 cells, as well as with the chirality shift induced by BMP2 (Fig.6). Previous study demonstrated that Pfn1 knockdown in human fibroblast shifts chirality from CW to CCW (Tee et al., 2023), suggesting that observed downregulation of Pfn1 in BMP2-induced C2C12 cell may indeed causatively shift the cell chirality from CW to CCW. In other words, our results are consistent with the view that lower Pfn1 expression in the presence of Actn1 favors CCW chirality (Tee et al., 2023).

3. The optimization of cell density for even distribution is noted, but the rationale remains unclear. Is monolayer formation essential for proper observation?

To address the reviewer's comments, we have added the following text to the discussion section and appendix S1 of the revised manuscript.

In discussion:

“The smaller the initial or final cell numbers on micropatterns, the greater the angular variability and the smaller the kurtosis of the orientation histograms (Appendix S1D and E). Therefore, even distribution of cells within the dish is important to ensure a sufficient number of cells on the micropattern for determination of collective cell chirality.”

In appendix S1:

“Dominant orientation data shown in Fig S1C (red, 5x10⁴ cell seeding density per dish) were further divided into four groups according to the initial (Fig S1D) or final (Fig S1E) number of cells in the micropatterns.”

In appendix S1 legend:

“(D) Orientation histograms of the four groups according to the initial cell number on each micropattern after seeding 5x10⁴ cells (Fig S1C, red). (E) Orientation histograms of the four groups according to the final cell number of on each micropattern after seeding 5x10⁴ cells (Fig S1C, red).”

4. The choice of C2C12 and MC3T3-E1 cells needs stronger biological rationale beyond their chirality differences. What controls were implemented to isolate chirality effects from other substrate/environmental influences? Would inclusion of additional cell types help validate method generalizability?

We chose C2C12 and MC3T3-E1 cells based on previous reports demonstrating that they exhibit opposite chirality (Wan et al 2011), and that C2C12 myoblasts can transdifferentiate into osteoblast-like cells when stimulated with BMP2 (Katagiri et al 1994). These reports prompted us to compare the chirality of C2C12 cells before and after BMP2 treatment with that of MC3T3-E1 cells thereby revealing the possible effect of induced cell differentiation on cell chirality.

In response to this reviewer, we performed additional experiments to include a new control, namely, MC3T3-E1 cells on vitronectin-coated micropatterns and analyzed chirality. Compared with fibronectin, vitronectin coating resulted in smaller dominant orientation of MC3T3-E1 cells, as was the case for C2C12 cells in terms of absolute angles. Since C2C12 and MC3T3-E1 cells exhibited opposite chirality on both vitronectin and fibronectin, it can be concluded that the two cell types exhibit different chirality regardless of the substrate. For the future study, we have started analyzing primary calvarial osteoblasts and bone marrow-derived osteoclast progenitors to see that this method is generalizable to primary cells.

We have added the following text to the introduction and results section of the revised manuscript.

In introduction:

“These reports prompted us to compare the chirality of C2C12 cells before and after BMP2 treatment with that of MC3T3-E1 cells thereby revealing the possible effect of induced cell differentiation on cell chirality.”

In results:

“Consistently, the average value of the dominant orientation of MC3T3-E1 cells was smaller on vitronectin than on fibronectin (Appendix S3A). Since the C2C12 and MC3T3-E1 cells exhibited opposite chirality on both vitronectin and fibronectin, it can be concluded that the two cell types exhibit different chirality regardless of the substrate.”

5. The manuscript contains inconsistent capitalization of figure panel labels (e.g., uppercase A/B/C vs. lowercase a/b/c).

The lowercase letters a, b, and c are not used as panel labels, but rather to indicate each step in the procedure. In response to the reviewer we added “Steps:” and changed capitalization from a, b, … and l to "i", "ii" … and "xii".

Reviewer #2: Manuscript: PONE-D-25-25096

Title: Use of a New Micropattern Tape Method to Detect Chirality shifts in differentiating C2C12 Cells

General Comments: Chirality is a property of cells demonstrating right and left behavior in terms of morphology and organization. The present paper develops a new method to pattern small regions of cells applied in rectangular micro patterns for chirality analysis. The authors develop a unique method utilizing a tape system which enables application of a range of extracellular matrix proteins while creating gap regions between the applied cell areas. They also have developed a sophisticated optical microscopic analysis system to determine orientation and a processing method for quantitation of images obtained. Using this method they look at the impact of an exogenous agent - bone morphogenic protein on alteration of chirality of two cell lines C2C 12 and MC3T3E1. They specifically show that the system can detect plasticity of cell chirality in vitro. The paper in general is well written is clear and describes the unique method. There are few areas which may be addressed to strengthen the paper

General issues to be discussed:

1. Provide a bit more in the discussion of other existing in vitro fabrication methods that other investigators are using and provide a bit more point by point detail of why your method is better.

To address the reviewer's comments, we have added the following text to the discussion section of the revised manuscript.

“There are several other methods for creating micropatterns, including PDMS stamping [5,22] and stencil methods [18]. The PDMS stamping method requires the creation of a mold. Molds can be made using cutting processes [23], 3D printing [24], or using photoresist [25]. The tape method has several advantages. It allows for the easy, stable fabrication of a large number of well-shaped micropatterns with a high yield. The tape itself is also simple to remove as opposed to a hardened resin mask outside the micropattern in the stencil method [18]. However, there are two limitations to this method. First, there is a size limitation. Designs of the micropattern with a scale of a few micrometers are too small to be excised out. Second, the inner enclosed part within a micropattern, such as a circle within a ring, is difficult to make. The inner part falls out during the cutting process of the tape.”

2. Please add detail as to the nature of the tape, its constituents, its composition and the nature of the adhesive.

As stated in the Methods section, the tape used is a commercially available product, the custom-designed IHC-Fencing Seal (CS CRIE). While the company does not disclose the tape's exact composition, it does accept custom design orders.

https://www.cscrie-eng.com/company/

3. Please add detail as to the uncoated film bottom dish. What is this material? Is this available for other investigators. In general, both of these details in points #2 and #3 above are important if a method is going to be more universally applied by others.

As mentioned in the Methods section, the uncoated, film-bottom dish (81151, ibidi) was purchased from ibidi. This product is publicly available for purchase. The material is referred to as "ibidi polymer," and its specific composition is proprietary information.

4. For figure 5 where there is reference to a range of extracellular matrix proteins please clarify if this involved the same general coating technique of the dish/microislands or is this different?? This is confusing to the reader.

To examine the effect of extracellular matrix proteins on cell differentiation, we conducted cell differentiation experiments using several different coatings (Figure 5). The experiment was conducted using a standard plastic cell culture dish and conventional coating methods.

The detailed experimental methods are described in the Methods section under the heading "Osteoblastic differentiation and evaluation of alkaline phosphatase activity."

5. In the discussion at the beginning in the first sentence it states, “cell adhesive fibronectin patterns”. Related to point #4 if this is only for fibronectin that is fine. However, if other adhesive proteins are being utilized here as well this is confusing to the reader and needs to be clarified.

In response to the reviewer's comment, we revised the manuscript to clarify that the tape method can be used for micropatterning with not only fibronectin, but also vitronectin and other coating proteins.

“In this study, we developed a micropattern tape method that simplifies fabrication of cell adhesive coating patterns, allowing high-throughput analysis of cell chirality.”

6. On page 18 lines 390 through 392 need to be clarified. Specifically, the issue of what is being adhered to what, and to prevent adhesion to uncoated areas is unclear to the reader - please explain.

Accordingly, we have revised the manuscript as follows:

“ Proper adhesion of the tape to the dish surface was ensured by tightly pressing the tape to prevent adhesion of coating proteins (fibronectin or vitronectin) to undesired areas. “

7. In the manuscript the word “holes” was used to designate the gap areas. Would use a different term such as uncoated area or region. For most readers holes will convey a mental image of something round rather than the rectangular regions you actually create.

Accordingly, we have revised the manuscript as follows:

“The design of our micropattern tape incorporated 453 rectangular cutouts arranged in a circle of 19 mm diameter, allowing data acquisition from a large number of rectangles (Fig 1B).”

8. would provide a bit more in the discussion re the sensitivity of this techniques and the optical/image analysis system you developed. How sensitive is this? How accurate? How does this compare to other systems.

To address the reviewer's comments, we have added the following text to the discussion section of the revised manuscript.

“This tape method enables a larger sample size per experiment, leading to a more robust and quantitative evaluation. In our hands, methods such as PDMS stamping were less stable, and required tedious exclusion of micropatterns with poor morphology. However, the tape method produces micropatterns with consistent, stable shapes, which provides at least comparable accuracy regardless of the amount of technical experience. Furthermore, our analysis system automates the process of cropping and preparing numerous micropatterns for analysis. Its accuracy depends on the use of the widely adopted ImageJ plugin, OrientationJ, for angle calculation [18,26].”

9. As to the biological findings of chirality changing with differentiation and cell aging or ECM effects – would expand on this a bit with discussions as to what is known in the literature and how do your findings conform or extend the body of knowledge.

To address the reviewer's comments, we have added the following text to the discussion section of the revised manuscript.

"Interestingly, others have reported a chirality shift from CCW to CW by differentiating mesenchymal stem cells to adipocytes [29], in contrast to CW to CCW shift from myoblasts to osteoblasts (Fig. 6B) . Changes in cell chirality may also occur in cell aging considering a reported association between aging and cytoskeleton organization in mesenchymal stem cells under self-renewal and osteogenic conditions [30]. Curiously, the chirality of an extracellular matrix may affect differentiation of mesenchymal stem cells to osteogenic and adipogenic cells possibly through cell chirality modulation [22].

The tape method with high density micropatterns reported here provides a versatile analytical tool allowing quantitative measurement of the cell chirality spectrum in terms of distribution of the dominant orientation. Even cells with the same handedness showed significant shifts in distribution of the dominant orientation (Fig 6A). Although established cell lines were used in this study, the tape method also facilitates chirality analysis of primary cells. Combining our tape method with in vivo chirality analysis during devel

---

## [Decision Letter · Decision Letter 1]

17 Nov 2025

Use of a new micropattern tape method to detect chirality shifts in differentiating C2C12 cells

PONE-D-25-25096R1

Dear Dr. Kawaai,

We’re pleased to inform you that your manuscript has been judged scientifically suitable for publication and will be formally accepted for publication once it meets all outstanding technical requirements.

Kind regards,

Baeckkyoung Sung, Ph.D.

Academic Editor

PLOS ONE

Additional Editor Comments (optional):

The revised manuscript has properly addressed the concerns raised by the reviewers.

Reviewers' comments:

Reviewer's Responses to Questions

**Comments to the Author**

Reviewer #1: All comments have been addressed

2. Is the manuscript technically sound, and do the data support the conclusions?

Reviewer #1: Yes

3. Has the statistical analysis been performed appropriately and rigorously?

Reviewer #1: Yes

4. Have the authors made all data underlying the findings in their manuscript fully available?

Reviewer #1: Yes

5. Is the manuscript presented in an intelligible fashion and written in standard English?

Reviewer #1: Yes

Reviewer #1: Having examined the authors' responses and the revised manuscript, we find that all of our previous concerns have been addressed. The revisions have adequately improved the quality of the work. The paper in its current form is satisfactory.

**Do you want your identity to be public for this peer review?** For information about this choice, including consent withdrawal, please see our Privacy Policy

Reviewer #1: No

---

## [Editor Report · Acceptance letter]

PONE-D-25-25096R1

PLOS ONE

Dear Dr. Kawaai,

I'm pleased to inform you that your manuscript has been deemed suitable for publication in PLOS ONE. Congratulations! Your manuscript is now being handed over to our production team.

Kind regards,

on behalf of

Dr. Baeckkyoung Sung

Academic Editor

PLOS ONE